# A Study on the Direct Esterification of Monoalkylphosphates and Dialkylphosphates; The Conversion of the Latter Species to Trialkylphosphates by Alkylating Esterification [note 1]

**DOI:** 10.3390/molecules27154674

**Published:** 2022-07-22

**Authors:** Péter Ábrányi-Balogh, Nikoletta Harsági, László Drahos, György Keglevich

**Affiliations:** 1Medicinal Chemistry Research Group, Research Centre for Natural Sciences, 1117 Budapest, Hungary; abranyi-balogh.peter@ttk.hu; 2Department of Organic Chemistry and Technology, Budapest University of Technology and Economics, 1521 Budapest, Hungary; harsagi.nikoletta@vbk.bme.hu; 3MS Proteomics Research Group, Research Centre for Natural Sciences, 1117 Budapest, Hungary; drahos.laszlo@ttk.hu

**Keywords:** P-ester acid, direct esterification, selectivity, alkylating esterification, energetics, mechanism, theoretical calculations, green method

## Abstract

The microwave (MW)-assisted direct esterification of certain P-acids is a green method. Quantum chemical calculations revealed that the activation enthalpy (Δ*H*^#^) for the exothermic monoalkylphosphate → dialkylphosphate transformation was on the average 156.6 kJ mol^−1^, while Δ*H*^#^ for the dialkylphosphate → trialkylphosphate conversion was somewhat higher, 171.2 kJ mol^−1^, and the energetics of the elemental steps of this esterification was less favorable. The direct monoesterification may be performed on MW irradiation in the presence of a suitable ionic liquid additive. However, the second step, with the less favorable energetics as a whole, could not be promoted by MWs. Hence, dialkylphosphates had to be converted to triesters by another method that was alkylation. In this way, it was also possible to synthesize triesters with different alkyl groups. Eventually a green, P-chloride free MW-promoted two-step method was elaborated for the synthesis of phosphate triesters.

## 1. Introduction

Microwave irradiation is a useful tool in promoting organic chemical reactions [1,2,3,4,5,6,7]. On the one hand, the transformations become faster with MW assistance; on the other hand, the conversions are more selective. Overall, the reactions can be accomplished in a more efficient way [8]. Another value is when MW irradiation substitutes catalysts [9,10,11] or allows the simplification of catalyst systems [12]. The greatest advantage of MWs is when reactions reluctant to conventional heating take place with irradiation [13].

An interesting discipline is the synthesis of P-esters, such as phosphinates, phosphonates and phosphates. The traditional way is to start from P-chlorides (phosphinic chloride, phosphonic dichloride and phosphorus oxychloride), and to react them with alcohols or phenols in the presence of a base [14,15]. However, these transformations require cost-meaning P-chlorides, and are not atomic efficient. We were successful in developing an MW-assisted, [bmim][PF_6_]-promoted method for the direct esterification of a series of phosphinic acids (Figure 1A) [16]. Phosphonic acids could also be converted to monoalkylphosphonates in a similar way using [bmim][BF_4_] (Figure 1B) [17]. The series is complete if the monoalkylphosphate → dialkylphosphate transformation is also considered (Figure 1C) [18]. The MW-assisted direct esterification of P-acids is an important method, as, in this way, the use of P-chlorides can be avoided. Hence, costs may be saved, and the formation of hydrochloric acid may be avoided.

In this paper, we wished to evaluate the energetics of the **V** → **VI** transformation, and that of the **VI** → (R^2^O)_3_P(O) conversion. Moreover, it was our purpose to elaborate the esterification of dialkylphosphates **VI**.

## 2. Results and Discussion

### 2.1. MW-Assisted Direct Esterification of Monoalkylphosphates

The monoalkylphosphates (**1a**–**d**) selected underwent esterification in reaction with the corresponding alcohol used in 15-fold quantity in the presence of 10% of [bmim][BF_4_] as the catalyst at 175/200 °C under MW irradiation. Our preliminary results were useful to find the optimum conditions [18]. The dialkylphosphates (**2a**–**d**) were obtained selectively, in yields of 83–87% after chromatography (Table 1). The main role of the ionic liquid additive is to act as an MW absorber in the reaction mixture [17]. Our earlier experiences showed that in the absence of an ionic liquid additive, the efficiency of the esterifications was significantly lower, when compared to the case when 10% of the catalyst was applied. The difference may have amounted to 80% [17].

The dialkylphosphates (**2**) could not be converted to the triesters (**3**) in a similar fashion.

### 2.2. Theoretical Calculations on the Energetics and Mechanism of the Monoalkylphosphate → Dialkylphosphate → Trialkylphosphate Transformation

We analyzed the energetics of the direct esterification of phosphates (R = Et, Bu) with the corresponding alcohols (EtOH, BuOH) using DFT computations at the M062X/6–311+G (d,p) level of theory considering the solvent effect (SMD implicit solvent model) of the corresponding alcohol and 473 K as the temperature (Figure 2, Table 2, Figure 1). Based on our previous model [17,19], we proposed a reaction complex containing three alcohol molecules and two phosphonic acid units, where one alcohol molecule acts as the reagent in the esterification. The other diester acid and ROH species in the reaction complex participated in the proton transfer chain supporting the establishment of the new P–O bond, and hence the formation of the diester acid, along with the departure of a water molecule.

Considering the difference in the energetics between the starting monoalkylphosphates and the final dialkylphosphates, the reaction may be regarded as slightly exothermic, supported by an enthalpy value of Δ*H* = −10.4 kJ mol^−1^ for the ethyl, and ΔH = −16.3 kJ mol^−1^ for the butyl substituted case. While the formation of the reaction complex (**4**) required a ca. 160 kJ mol^−1^ Gibbs free energy (Δ*G*) investment for both cases (see Appendix A) that was the consequence of the entropy increase during the complex formation, there was a significant enthalpy gain (Δ*H* = −151.6 kJ mol^−1^ for the ethyl and −160.9 kJ mol^−1^ for the butyl instance). As shown in **TS1** (Δ*H*^#^ = 94.7 kJ mol^−1^ and 93.0 kJ mol^−1^, respectively), the next step of the reaction was the attack of the alcohol on the phosphorus atom of the P=O moiety leading to intermediate **5**. The following step was the elimination of a water molecule via **TS2** (Δ*H*^#^ = 158.8 kJ mol^−1^ and 154.3 kJ mol^−1^, respectively) yielding product complex **6**. The difference in the relative enthalpy (ΔΔ*H*) of **6** and **4** was 1.2 kJ mol^−1^ and −4.4 kJ mol^−1^ for the two cases. At the same time, the gain in Δ*G* was larger (−22.6 kJ mol^−1^ and −11.3 kJ mol^−1^, respectively). The disruption of complex **6** was driven by Δ*H =* −10.4 kJ mol^−1^ and −16.3 kJ mol^−1^ (as well as by Δ*G* = −4.9 kJ mol^−1^ and −7.8 kJ mol^−1^) for the ethyl and butyl substituted case, respectively. The whole sequence was just slightly exothermic requiring a high activation energy investment mainly due to the large entropy that needed to be overcome. This supports the need for harsh experimental conditions ensured by the MW irradiation at 200 °C.

Investigating the transformation of diethyl and dibutylphosphate to triethyl and tributylphosphate, we found that the total process was somewhat less exothermic (Δ*H* = −7.6 kJ mol^−1^ and −13.7 kJ mol^−1^, respectively). The formation of the reaction complex was also less advantageous (Δ*H* = −143.3 kJ mol^−1^ and −144.1 kJ mol^−1^) as compared to the monoalkyl → dialkyl transformation. Moreover, both following steps required a higher activation enthalpy (for **TS1** Δ*H*^#^ = 98.2 kJ mol^−1^ and 108.1 kJ mol^−1^, respectively, and for **TS2**, 173.7 kJ mol^−1^ and 168.7 kJ mol^−1^, respectively). Finally, the stabilization of **TS2** to intermediate **6** was less advantageous, and significantly lower enthalpy gains (Δ*H* = −121.0 kJ mol^−1^ and −107.8 kJ mol^−1^) could be observed, suggesting in total an endothermic **4** → **6** transformation (ΔΔ*H* = 22.3 kJ mol^−1^ and 36.3 kJ mol^−1^, respectively, and ΔΔ*G* = 23.0 kJ mol^−1^ and 34.1 kJ mol^−1^, respectively).

### 2.3. MW-Assisted Alkylation of Dialkylphosphates

We saw that the dialkylphosphates (**2**) resisted undergoing further esterification to the triesters (**3**) that is due to the high barrier of the activation enthalpy. Hence, the conversion of diesters **2** to trialkylphosphates **3** had to be carried out by another method, by alkylating esterification. This was realized by applying the corresponding alkyl halides (bromobutane, bromopentane, bromopropane and iodoethane) together with triethylamine as the base in toluene at 135 °C on MW irradiation. Again, our earlier results were useful in finding the optimum conditions [18]. The results are collected in Table 3. It can be seen that the trialkylphosphates were obtained in 84–86% yields after the chromatography.

We thought that the alkylating esterification may be also suitable for the preparation of trialkylphosphates with different alkyl groups. Dibutylphosphate **2a** was reacted, as shown above, with a few haloalkanes. The results are shown in Table 4. One may suspect that the difference of the two conversions covers the side-reactions. Indeed, LC–MS pointed out the presence of HOP(O)(OR)OBu, HOP(O)(OR)_2_, (RO)_2_P(O)OBu and (RO)_3_P(O) by-products as well, during the reaction of diester **2a** with haloalcanes. Their formation is not completely clear, and interconversions to the effect of the Et_3_N-HBr salt under MW irradiation are assumed.

Dipentylphosphate **2b** was also subjected to alkylations. The experimental data are collected in Table 5. In this case, the proportion of the by-products was somewhat higher.

The trialkylphosphates with different alkyl groups (**7** and **8**) synthesized by us were mostly new compounds. A few of them were described but were not fully characterized. We characterized all “mixed” derivatives by ^31^P, ^13^C and ^1^H NMR data, as well as HRMS.

## 3. Materials and Methods

### 3.1. General Information

The ^31^P, ^13^C and ^1^H NMR spectra were taken on a Bruker DRX-500 spectrometer operating at 202.4, 125.7 and 500 MHz, respectively. LC–MS measurements were performed with an Agilent 1200 liquid chromatography system coupled with a 6130 quadrupole mass spectrometer equipped with an ESI ion source (Agilent Technologies, Palo Alto, CA, USA). The MW-assisted esterifications were carried out in a CEM Discover microwave reactor equipped with a stirrer and a pressure controller using a 50–100 W irradiation.

The composition of the reaction mixtures was determined by the integration of the areas under the corresponding peaks of the starting material and product in the ^31^P NMR spectra. As the ^31^P NMR signals separated better in DMSO-D_6_, this solvent was used during the analysis of the mixtures.

### 3.2. The Direct Esterification of Monoalkylphosphates (**1a**–**d**)

A mixture of 0.79 mmol monoalkylphosphate (**1a**: 0.12 g; **1b**: 0.13 g; **1c**: 0.11 g; **1d**: 0.10 g) (prepared as described above), 11.9 mmol of alcohol (ethanol: 0.69 mL; propanol: 0.89 mL; butanol: 1.08 mL; pentanol: 1.30 mL) and 15 µL (0.079 mmol) of [bmim][BF_4_] was irradiated in the MW reactor at 175–200 °C for 2–4.5 h (Table 1). The crude mixture obtained on evaporation was purified by chromatography using a silica-gel layer of 20 cm, and ethyl acetate as the eluent to furnish dialkylphosphates (**2a**–**d**) as colorless oils. For identification of the dialkylphosphates, see Table 6.

### 3.3. The Alkylating Esterification of Dialkylphosphates (**2a**–**2d**)

A mixture of 1.4 mmol (**2a**: 0.30 g, **2b**: 0.31 g, **2d**: 0.22 g) dialkylphosphate, 1.8 mmol (EtBr: 0.14 mL, PrBr: 0.17 mL, ^i^PrBr: 0.17 mL, BuBr: 0.20 mL, PentBr: 0.22 mL) of alkyl bromide and 0.22 mL (1.6 mmol) of triethylamine in 1 mL of toluene was stirred under MW conditions for 2–5 h at 135–150 °C (Table 3, Table 4 and Table 5). The crude mixtures obtained after filtration and evaporation were purified by column chromatography using a silica gel layer of 20 cm and ethyl acetate as the eluent to afford the corresponding trialkylphosphates (**3a**–**d, 7a**–**d** and **8a**–**d**) as colorless oils. For the identification of the known trialkylphosphates, see Table 7.

### 3.4. Characterization of New Trialkylphosphates (**7a**–**d** and **8a**–**d**)

#### 3.4.1. Dibutyl-ethylphosphate (**7a**)

^31^P NMR (202.4 MHz, CDCl_3_) δ: −0.79, δ_P_ [20] (CDCl_3_): 0.75; ^13^C NMR (125.7 MHz, CDCl_3_) δ: 13.5 (s, 2CH_3_), 16.1 (d, *J =* 6.7, CH_3_), 18.6 (s, 2CH_2_), 32.2 (d, *J =* 6.8, 2CH_2_), 63.5 (d, *J =* 5.9, OCH_2_), 67.3 (d, *J =* 6.0, 2OCH_2_); ^1^H NMR (500 MHz, CDCl_3_) δ: 0.95 (t, *J =* 7.4, 6H, 2CH_3_), 1.33–1.46 (m, 7H, CH_3_, 2CH_2_), 1.63–1.72 (m, 4H, 2CH_2_), 4.02–4.17 (m, 6H, 3OCH_2_), δ_H_ [21] (CDCl_3_): 0.91 (t, 6H, *J =* 6.5), 1.25 (t, 3H, *J =* 5.8), 1.35–1.88 (m, 8H), 3.80–4.30 (m, 6H). [M+Na]^+^_found_: 261.1227, [M+Na]^+^_calculated_: 261.1232.

#### 3.4.2. Dibutyl-propylphosphate (**7b**)

^31^P NMR (202.4 MHz, CDCl_3_) δ: −0.75, ^13^C NMR (125.7 MHz, CDCl_3_) δ: 9.9 (s, CH_3_), 13.4 (s, 2 CH_3_), 18.6 (s, 2CH_2_), 23.6 (d, *J =* 6.9, CH_2_), 32.2 (d, *J =* 6.8, 2CH_2_), 67.2 (d, *J =* 6.3, 2OCH_2_), 69.0 (d, *J =* 6.0, OCH_2_); ^1^H NMR (500 MHz, CDCl_3_) δ: 0.94 (dt, *J =* 13.4, *J =* 7.4, 9H, 3CH_3_), 1.36–1.44 (m, 4H, 2CH_2_), 1.62–1.71 (m, 6H, 3CH_2_), 3.96–4.04 (m, 6H, 3OCH_2_). [M+Na]^+^_found_: 275.1390, [M+Na]^+^_calculated_: 275.1388.

#### 3.4.3. Dibutyl-isopropylphosphate (**7c**)

^31^P NMR (202.4 MHz, CDCl_3_) δ: 0.50, δ_P_ [20] (CDCl_3_): 0.60; ^13^C NMR (125.7 MHz, CDCl_3_) δ: 13.6 (s, 2 CH_3_), 18.7 (s, 2 CH_2_), 23.6 (d, *J =* 5.0, 2CH_3_), 32.3 (d, *J =* 7.0, 2CH_2_), 67.2 (d, *J =* 6.2, 2OCH_2_), 72.3 (d, *J =* 5.8, OCH); ^1^H NMR (500 MHz, CDCl_3_) δ: 0.93 (t, *J =* 7.4, 6H, 2CH_3_,), 1.33 (d, *J =* 6.2, 6H, 2CH_3_), 1.37–1.46 (m, 4H, 2CH_2_), 1.63–1.69 (m, 4H, 2CH_2_), 3.99–4.05 (m, 4H, 2OCH_2_), 4.60–4.66 (m, 1H, OCH), δ_H_ [21] (CDCl_3_): 0.65–1.88 (m, 23H), 3.63 (d, 2H, *J =* 5.3), 3.99 (dt, 4H, *J =* 6.5, *J =* 7.5). [M+Na]^+^_found_: 275.1386, [M+Na]^+^_calculated_: 275.1388.

#### 3.4.4. Dibutyl-pentylphosphate (**7d**)

^31^P NMR (202.4 MHz, CDCl_3_) δ: −0.68; ^13^C NMR (125.7 MHz, CDCl_3_) δ: 13.5 (s, 2CH_3_), 13.9 (s, CH_3_), 18.6 (s, 2 CH_2_), 22.2 (s, CH_2_), 27.5 (s, CH_2_), 29.9 (d, *J =* 6.8, CH_2_), 32.3 (d, *J =* 6.9, 2 CH_2_), 67.3 (d, *J =* 6.2, 2 OCH_2_), 67.6 (d, *J =* 6.2, OCH_2_); ^1^H NMR (500 MHz, CDCl_3_) δ: 0.88–0.94 (m, 9H, 3CH_3_), 1.31–1.36 (m, 4H, 2CH_2_), 1.38–1.44 (m, 4H, 2CH_2_), 1.62–1.69 (m, 6H, 3CH_2_), 3.99–4.04 (m, 6H, 3OCH_2_). [M+Na]^+^_found_: 303.1699, [M+Na]^+^_calculated_: 303.1701.

#### 3.4.5. Dipentyl-ethylphosphate (**8a**)

^31^P NMR (202.4 MHz, CDCl_3_) δ: −0.75; ^13^C NMR (125.7 MHz, CDCl_3_) δ: 13.9 (s, 2CH_3_), 16.1 (d, *J =* 7.0, CH_3_), 22.2 (s, 2CH_2_), 27.6 (s, 2CH_2_), 30.0 (d, *J =* 7.0, 2CH_2_), 63.6 (d, *J =* 5.9, OCH_2_), 67.7 (d, *J =* 6.1, 2OCH_2_); ^1^H NMR (500 MHz, CDCl_3_) δ: 0.91 (t, *J =* 6.9, 6H, 2CH_3_), 1.33–1.38 (m, 11H, 4CH_2_, CH_3_), 1.66–1.71 (m, 4H, 2CH_2_), 3.99–4.05 (m, 4H, 2OCH_2_), 4.08–4.14 (m, 2H, OCH_2_). [M+Na]^+^_found_: 289.1544, [M+Na]^+^_calculated_: 289.1545.

#### 3.4.6. Dipentyl-propylphosphate (**8b**)

^31^P NMR (202.4 MHz, CDCl_3_) δ: −0.70; ^13^C NMR (125.7 MHz, CDCl_3_) δ: 10.0 (s, CH_3_), 13.9 (s, 2CH_3_), 22.2 (s, 2CH_2_), 23.6 (d, *J =* 6.9, CH_2_), 27.6 (s, 2CH_2_), 30.0 (d, *J =* 6.8, 2CH_2_), 67.6 (d, *J =* 6.0, 2OCH_2_), 69.1 (d, *J =* 6.0, OCH_2_); ^1^H NMR (500 MHz, CDCl_3_) δ: 0.91 (t, *J =* 7.1, 6H, 2CH_3_), 0.97 (t, *J =* 7.4, 3H, CH_3_), 1.32–1.38 (m, 8H, 4CH_2_), 1.66–1.74 (m, 6H, 3CH_2_), 3.97–4.05 (m, 6H, 3OCH_2_). [M+Na]^+^_found_: 303.1701, [M+Na]^+^_calculated_: 303.1701.

#### 3.4.7. Dipentyl-isopropylphosphate (**8c**)

^31^P NMR (202.4 MHz, CDCl_3_) δ: −1.62; ^13^C NMR (125.7 MHz, CDCl_3_) δ: 13.9 (s, 2CH_3_), 22.2 (s, 2CH_2_), 23.6 (d, *J =* 5.0, 2CH_3_), 27.6 (s, 2CH_2_), 30.0 (d, *J =* 7.1, 2CH_2_), 67.5 (d, *J =* 6.2, 2OCH_2_), 72.3 (d, *J =* 5.9, OCH); ^1^H NMR (500 MHz, CDCl_3_) δ: 0.91 (t, *J =* 6.9, 6H, 2CH_3_), 1.26–1.40 (m, 14H, 4CH_2_, 2CH_3_), 1.66–1.72 (m, 4H, 2CH_2_), 4.00–4.05 (m, 4H, 2OCH_2_), 4.61–4.68 (m, 1H, OCH). [M+Na]^+^_found_: 303.1703, [M+Na]^+^_calculated_: 303.1701.

#### 3.4.8. Dipentyl-butylphosphate (**8d**)

^31^P NMR (202.4 MHz, CDCl_3_) δ: −0.62; ^13^C NMR (125.7 MHz, CDCl_3_) δ: 13.6 (s, CH_3_), 13.9 (s, 2 CH_3_), 18.7 (s, CH_2_), 22.2 (s, 2CH_2_), 27.6 (s, 2CH_2_), 30.0 (d, *J =* 6.8, 2CH_2_), 32.2 (d, *J =* 6.8, CH_2_), 67.4 (d, *J =* 6.1, OCH_2_), 67.7 (d, *J =* 6.1, 2OCH_2_); ^1^H NMR (500 MHz, CDCl_3_) δ: 0.93 (t, *J =* 7.2, 6H, 2CH_3_), 0.96 (t, *J =* 7.7, 3H, CH_3_), 1.33–1.40 (m, 8H, 4CH_2_), 1.41–1.46 (m, 2H, CH_2_), 1.67–1.72 (m, 6H, 3CH_2_), 4.03–4.08 (m, 6H, 3OCH_2_). [M+Na]^+^_found_: 317.1857, [M+Na]^+^_calculated_: 317.1858.

For the NMR spectra of the products, see the Appendix A.

### 3.5. Theoretical Calculations

DFT computations at the M062X/6–311+G (d,p) level of theory were performed considering the solvent effect of the corresponding alcohol using the SMD solvent model with the Gaussian 09 program package [21,22,23]. The geometries of the molecules were optimized in all cases, and frequency calculations were also performed to ensure that the structures were in a local minimum or in a saddle point. The conformations of the reported structures were determined by conformational analysis. The solution-phase enthalpies and Gibbs free energies were obtained by frequency calculations as well. The H and G values obtained were given under 473 K, the corrected total energies of the molecules were taken into account. Entropic and thermal corrections were evaluated for isolated molecules using standard rigid rotor harmonic oscillator approximations, that is, the enthalpy and the Gibbs free energy were taken as the “sum of electronic and thermal free energies” printed in a Gaussian 09 vibrational frequency calculation. The standard state correction was taken into account. The transition states were optimized with the QST3 or the TS (Berny) method. The transition states were identified by having one imaginary frequency in the Hessian matrix, and IRC calculations were performed in order to prove that the transition states connected two corresponding minima.

For the details of the calculations, see the Appendix A.

## 4. Conclusions

An MW-assisted protocol was developed for the esterification of monoalkylphosphates. The first step was the chemoselective direct esterification in the presence of [bmim][BF_4_] as the catalyst. The second step was an alkylation esterification. Even phosphoric triesters with different alkyl groups were prepared. Additionally, quantum chemical computations showed that the activation enthalpy was high (on average 156.6 kJ mol^−1^) for the monoesterifications, and even higher for the diesterifications, which agreed with the observed experimental data. In addition, the determining effect of entropy was pointed out in the esterifications. It is also noted that regarding direct esterifications, the overall energetics for the formation of diesters was more favorable than that for the formation of the triesters. As a whole, a new method was developed for the preparation of phosphate triesters avoiding the use of P-chlorides as the starting materials. The first, direct MW-assisted esterification step may be regarded as “green”. The experimental data were supported by theoretical calculations.

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
