# Peer review of "A Study on the Direct Esterification of Monoalkylphosphates and Dialkylphosphates; The Conversion of the Latter Species to Trialkylphosphates by Alkylating Esterification†"

_molecules, 2022, doi:10.3390/molecules27154674_

Round 1
Reviewer 1 Report
Recommendation: Minor revision.
Microwave irradiation is a useful tool in promoting organic chemical reactions. In this manuscript, the authors report a MW-assisted protocol for the esterification of monoalkylphosphates. In addition, quantum chemical computations showed that the activation enthalpy was high for the monoesterifications, and even higher for the diesterifications that was in accord with the observed experimental data. And eventually, they used a green, P-chloride-free MW-promoted two-step method to achieve phosphate triesters. The experimental design was reasonable and the data were basically support the statements and the conclusion. Therefore, I recommend the acceptance of the manuscript after suitable revision.
1. Please check the Scheme 2. “MeOH”, should it be “ROH”?
Author Response
Dear Referee 1:
Thanks for the favorable opinion.
You wrote:
"Therefore, I recommend the acceptance of the manuscript after suitable revision. Please check the Scheme 2. “MeOH”, should it be “ROH”?"
No, on the arrow the two "MeOH" molecules are correct, as these species take only place in establishing the molecular network (associate). It was a simplification for the theoretical calculations not to apply larger ROH molecules in the network. Only the esterifying species was ROH. At the same time, you are right that Scheme 2 was not correct, as in the complex structures only ROH molecules appeared and not the simplifying MeOH species. This was corrected.
Thanks for spotting this mistake and supporting publication.
György Keglevich
Reviewer 2 Report
See attached file

Author Response
Dear Referee 2:
Thanks for your extremely useful remarks:
"... However, I have some comments and questions that need clarification:
Introduction:
- it is worth describing (2-4 sentences) the significance of the discussed transformations."
This was done on p. 2, above Scheme 1: "The MW-assisted direct esterification of P-acids is an important method, as, in this way, the use of P-chlorides can be avoided. Hence, costs may be saved, and the formation of hydrochloric acid may be avoided."
"- scheme 1 - please check if everything is correct - in the literature, you described mainly [bmim][PF6] (see Ref [18] and [20])"
Indeed, in Scheme 1/A (bmim)(PF6) is the correct agent. This was corrected.
"Results and discussion:
- In this article, the Authors do not explain the role of the ionic liquid [bmim][BF4], there is also no description of the reaction without the use of [bmim][BF4], which seems to be crucial for comparative purposes (an explanation and/or reference/citation will be useful).
The role if the IL was explained on p. 2, above Table 1: "The main role of the ionic liquid additive is to act as a MW absorber in the reaction mixture [17]. "
The need for an IL was justified in the following manner: "Our earlier experiences showed that in the absence of an ionic liquid additive, the efficiency of the esterifications was significantly lower, as compared to the case, when 10% of the catalyst was applied. The difference may have amounted to 80% [17]."
"- table 1 - how the reaction conditions were selected? (a description or reference/citation will be useful)"
It was inserted: "Our preliminary results were useful to find the optimum conditions [18]."
"- scheme 2 it should be 2 ROH instead of 2 MeOH (twice) see also Table S2 (SI)"
No! On the arrow the two "MeOH" molecules are correct, as these species take only place in establishing the molecular network (associate). It was a simplification for the theoretical calculations not to apply larger ROH molecules in the network. Only the esterifying species was ROH. At the same time, you are right that Scheme 2 was not correct, as in the complex structures only ROH molecules appeared and not the simplifying MeOH species. This was corrected. Consequently, the treatment in the Suppl. Info was correct, and we left MeOH.
"- table 3 - how the reaction conditions were selected? (a description or reference/citation will be useful)"
We inserted: "Again, our earlier results were useful in finding the optimum conditions [18]. "
"Materials and methods:
General information: What does it mean: The 31P NMR spectra were taken on a Bruker DRX-500 spectrometer operating at 202.4, 125.7 and 300 MHz, espectively."
Again you spotted a mistake. This was corrected as "The 31P, 13C, 1H NMR spectra were taken on a Bruker DRX-500 spectrometer operating at 202.4, 125.7 and 500 MHz, respectively"
"NMR descriptions:
compound 7d - the description of the 13C NMR spectrum does not match the attached spectrum"
Sorry for the mistake, this was acopy-paste problem. The correct NMR was inseretd insetad of the false data.
"compound 8a; line 215: should be 11H instead of 1H (1.33-1.38 m)"
Of course, this was also corrected.
"Conclusions (it should be the fourth section not the fifth):"
Corrected.
"- please check the given value of the enthalpy of activation (line 258; is it related to the first stage of esterification?)"
156.6 belongs to the 2. stage of the first esterification, and this value is the average of 158.9 and 154.3. See the original diagram (figure 1)!
"References:
- there are a lot of self-citations here (about 50%) - I suggest reducing this level a bit (e.g. references 8-15)"
OK, original references 8 and 12 were removed. Moreover, ref-s 19 and 23 (in the revised version 17) were the same. So 3 self-citations were eliminated.
"Supporting information:
- please add the frequencies at which the 31P, 1H, and 13C NMR spectra were recorded."
This was done.
"By the way, I am not entirely convinced whether this method (the second stage) can be classified as a green method (this phrase is much overused today). Could you justify it somehow (Pchlorides are not used here, but two steps have to be performed and the second one requires solvent (aromatic compound), excess base, and excess alkyl halide)."
This is true. For this the conclusionswere modified ("green" was replaced by "new", and the stress was put on the 1. step): "In whole, a new method was developed for the preparation of phosphate triesters avoiding the use of P-chlorides as the starting materials. The first, direct MW-assisted esterification step may be regarded “green"."
All corrections were highlighted in yellow.
Thanks again for the very careful evaluation of this perfect senior referee.
Round 2
Reviewer 2 Report
All comments and doubts were taken into account or explained by the Authors. Thank you. This is a very interesting article.
All the best